# Limitations of Foot-Worn Sensors for Assessing Running Power

**DOI:** 10.3390/s21154952

**Published:** 2021-07-21

**Authors:** Tobias Baumgartner, Steffen Held, Stefanie Klatt, Lars Donath

**Affiliations:** 1Institute of Exercise Training and Sport Informatics, Department of Cognitive and Team/Racket Sport Research, German Sport University Cologne, 50933 Cologne, Germany; t.baumgartner@dshs-koeln.de (T.B.); s.klatt@dshs-koeln.de (S.K.); 2Department of Intervention Research in Exercise Training, German Sport University Cologne, 50933 Cologne, Germany; s.held@dshs-koeln.de; 3School of Sport and Health Sciences, University of Brighton, Eastbourne BN20 7SR, UK

**Keywords:** accelerometer, running power, running economy, Stryd, metabolic cost of transportation

## Abstract

Running power as measured by foot-worn sensors is considered to be associated with the metabolic cost of running. In this study, we show that running economy needs to be taken into account when deriving metabolic cost from accelerometer data. We administered an experiment in which 32 experienced participants (age = 28 ± 7 years, weekly running distance = 51 ± 24 km) ran at a constant speed with modified spatiotemporal gait characteristics (stride length, ground contact time, use of arms). We recorded both their metabolic costs of transportation, as well as running power, as measured by a Stryd sensor. Purposely varying the running style impacts the running economy and leads to significant differences in the metabolic cost of running (*p* < 0.01). At the same time, the expected rise in running power does not follow this change, and there is a significant difference in the relation between metabolic cost and power (*p* < 0.001). These results stand in contrast to the previously reported link between metabolic and mechanical running characteristics estimated by foot-worn sensors. This casts doubt on the feasibility of measuring running power in the field, as well as using it as a training signal.

## 1. Introduction

In the last decade, foot-worn sensors to assess and meaningfully analyze running metrics (e.g., step frequency, stride length, ground contact time) have gained increased attention and popularity [1,2,3,4,5]. These sensors are meant to improve laboratory and in-field testing and training by delivering key performance data. The data derived from foot-worn sensors comprise spatiotemporal running parameters, including running power [1,6]. These devices are designed to be independent of factors such as slope, wind resistance, or fatigue. There has been significant prior research related to the use of foot-worn accelerometers for different purposes: they have been used for and are known to correlate with ground speed [7], running economy [8], and running power [6,9].

In this research, we aim to clarify whether it is possible to reliably measure running power using foot-worn sensors and use these data to objectively quantify effort. Being able to objectively evaluate and compare training effort are considered valuable to improve training programming and progression [10]. For example, two runners may cover a certain distance in the same duration, yet they experience different levels of exhaustion or underlying energy costs of running. This is mainly due to a complex interplay among the central and peripheral properties of cardiocirculatory, ventilatory, metabolic, and psychological capacities [11].

In practice, energy demands are measured as the rate of oxygen consumption normalized to body weight, specified as relative oxygen consumption $\dot{V}O_{2}$ [mL·min${}^{-1}\cdot$kg${}^{-1}$] [12,13]. A reliable assessment of such exchange data via spiroergometric systems is applicable to in-field, as well as laboratory conditions. However, these systems can be considered a bit cumbersome and, therefore, require trained professionals to assess and interpret the data.

Thus, the direct assessment of the energy output of an athlete without a spiroergometric mask seems to be promising from the perspective of athletes and coaches. In cycling, a common measure to evaluate absolute effort is power, measured in Watts. Power (=work per time) is often determined at a single point in the human–bike interface: the pedal. This deduction appears reasonable as the energy transferred from an athlete onto the street passes through this transmission point. Evidence shows that there is a very strong association between metabolic (measured by oxygen consumed) and mechanical power at the pedal in cycling (*r* = 0.97) [14]. With this background, it seems analogously plausible to measure running power also in a similar setup: The foot serves as the interface where the athlete’s effort is exerted. Force plates would then be the point of reference to measure ground reaction force (GRF) and derive power applied to the ground.

However, this setup would not be feasible in the field. Alternative solutions for GRF measurement using specialized insoles are still expensive and still need to be validated. Hence, numerous companies have introduced small devices that can be easily attached to the shoelaces. These compact and consumer-friendly devices contain accelerometers [15]. By using Newton’s second law, the force at the foot can be approximated via F = m·a.

A direct transfer of measurement parameters from cycling to running power, however, is problematic. The motion on a bike is very constrained, whereas runners move with more degrees of freedom [16]. Runners additionally use the muscle–tendon unit to regain and, thereby, conserve energy with each step [17]. These factors can be trained and altered and, therefore, affect the ratio between metabolic energy consumption and mechanical energy output. There are a number of factors influencing the relation between energy consumed and energy expended in propulsion [18,19], commonly referred to as running economy (RE) [16,20]. In this paper, we try to show that economy-related aspects need to be taken into account when considering running power.

The primary objective of this work, then, is to clarify the information meant to be gained from foot-worn sensors and to present the results of a simple experiment that manipulates athletes’ RE. We investigated the effects on energy intake ($\dot{V}O_{2}$) and power produced ($P_{W}$) when altering running parameters, such as stride length and step frequency. By keeping the running speed constant while simultaneously altering the efficiency of energy utilization, we created a gap between $\dot{V}O_{2}$ and $P_{W}$ as measured by a foot-worn sensor. We propose that we can alter $\dot{V}O_{2}$ significantly whilst the power output $P_{W}$ does not increase accordingly. This finding suggests that we are, in fact, not ready to measure metabolic running power in daily training with existing ready-to-use consumer products.

## 2. Methods

### 2.1. Participants and Study Design

A total of 32 moderate endurance-trained runners (males (n = 22), females (n = 10)) participated in this controlled crossover trial. The participants’ characteristics (mean SD) included: age = 28 ± 7 years, BMI = 21.6 ± 1.6, running $\dot{V}O_{2\max}$ = 58.6 ± 13.1 mL·min${}^{-1}$·kg${}^{-1}$, and weekly running distance = 51 ± 24 km over the prior four weeks. All participants had at least one year of distance running experience and were injury-free for at least the preceding three months. Participants chose their own footwear to best resemble their individual training and racing conditions. This study was approved by the local ethics committee (Ethics Commission German Sport University Cologne, Ethical Proposal No. 017/2021) and was in compliance with the declaration of Helsinki [21]. All participants were informed about the study design, and they signed informed consent for participation.

The entire test protocol was carried out in a single laboratory visit and was divided into two parts. First, each subject performed a combined incremental and ramp exercise test to determine their velocity at the aerobic threshold (AeT) [22]. For this, an incremental maximal oxygen uptake ($\dot{V}O_{2\max}$) treadmill test was conducted until objective exhaustion levels were reached [13]. Thereafter, following a 30 min rest, the participants ran another 25 min on the treadmill with varying predetermined spatiotemporal running parameters. Throughout all tests, the participants wore a heart rate transmitter chest strap and receiver/watch (Garmin, Olathe, KS, USA). Athletes refrained from intense exercise 48 h prior to the test.

### 2.2. Testing Procedures

#### 2.2.1. Combined Incremental and Ramp Exercise Test

In order to determine their $\dot{V}O_{2\max}$, as well as their AeT, the participants performed a combined incremental and ramp testing protocol. The initial speed of the subjects was set based on prior running experience and estimated 10 km race time at 2, 2.5, or 3 m·s${}^{-1}$. The combined protocol then consisted of the following steps: four 3 min stages (0.5 m·s${}^{-1}$ speed increase per stage, 30 s rest in between), immediately followed by 90 s at the same speed as the last stage, and then a ramp test (0.2 m·s${}^{-1}$ speed increase every 30 s) until the subjects reached volitional exhaustion [13]. In addition, capillary blood samples were taken from the earlobe of the participants for lactate analysis (EBIOplus; EKF Diagnostic Sales, Magdeburg, Germany) during the 30 s rest periods between stages and immediately after the ramp test.

$\dot{V}O_{2\max}$ data were collected, using a breath-by-breath spirometric system (Zan 600, Zan Messgeräte, Oberthulba, Germany). This spirometric system was calibrated prior to each test, following the manufacturer’s recommendations. The highest consecutive oxygen uptake values within 30 s during the final part were considered as $\dot{V}O_{2\max}$. $\dot{V}O_{2\max}$ and objective exhaustion were verified for each participant following the criteria by Midgley and colleagues [13]. All participants were verbally encouraged and motivated in the same way towards the end of $\dot{V}O_{2\max}$ testing, and they fulfilled objective exhaustion criteria (i.e., at least 4 out of 6 criteria). During a 30 min rest period between the all-out exhaustion test and the second part of the study, the athletes’ AeT was determined using the *minimum lactate equivalent* (L${}_{\min}$) method [22]. In order to do this, we fit an exponential function to the lactate measures and the velocities measured during the four stages. Thereafter, we determined L${}_{\min}$ as the minimum of the ratio between this function and the velocity, using custom-built Python functions.

Running power during all tests was measured using a Stryd sensor (Stryd Summit Powermeter, firmware 2.1.16; https://www.stryd.com/, Stryd, Inc., Boulder, CO, USA). This sensor was attached to an athlete’s shoe using a clip in the shoelaces and connected to a watch via Bluetooth. This device is lightweight (8.5 g), unobtrusive (4 cm length), and did not impact the athlete’s running form [15]. It was independently validated to strongly correlate with $\dot{V}O_{2}$ in a recent study by Cerezuela-Espejo et al. [6]. In their study, which included two foot-worn and three additional sensors, the Stryd sensor resulted in the strongest correlation between displayed power and the metabolic cost of running. The sensor was used in accordance with the manufacturer’s instructions, was reset between participants, and set to the weight and height of each athlete. It was attached to the bottom laces of the left foot and stayed fixed for a single subject between tests.

Data from the Stryd sensor and the heart rate monitor were sampled at 1 Hz and synchronized during the recording by a Garmin watch. Spirometric data were also recorded at a 1 Hz frequency and aligned manually during analysis. Potential errors due to misalignment were mitigated through data pooling.

#### 2.2.2. Modulation of Running Economy

In the second part of the study, the subjects ran for 25 min at a constant pace at their AeT, as calculated in the previous test. Before the test, each subject was instructed on the interrelation among step frequency, step length, and ground contact time while keeping a steady pace. They were also familiarized with the metronome (Weird Metronome, David Johnston, http://www.weirdmetronome.com, cf. [23]). After a brief warm-up period of 2 min, the next 3 min were used to determine the subject’s self-selected preferred step frequency (SF) at that pace. Next, at the same pace, the participants were instructed to perform four different variations to their running style for 3 min each. Between variations, the participants ran in their own unrestricted running form for 1 min. The four conditions in this experiment were as follows (order randomized between participants):SF+10%: an increase of the step frequency by 10%. This was prescribed using a digital metronome. Participants were also verbally encouraged and supported when the target frequency was not met;SF−10%: a decrease in step frequency by 10%. Again, a metronome was used to help the participants keep this running form. As additional mental help, the participants were instructed to lengthen their stride, as if they were gliding;GCT: reduction of ground contact time. The participants were instructed to reduce the time spent in contact with the ground by around 20 ms. They were told the GCT during their self-selected running and a target GCT for this variation. Participants were instructed regularly to either keep their step exactly as is or to try and decrease their GCT further. As a mental image, participants were encouraged to imagine the treadmill to be covered in hot coals;Arms: The participants were instructed to run without arm swing and, thus, without counterbalance to their running motion. The arms were either held above the head or in the neck in order to avoid effective use as a counterweight to rotational movement.

After completing the four conditions, the participants continued running in their own running style for another 5 min. We used the $\dot{V}O_{2}$ towards the end of this cool-down, to approximate fatigue. Assuming a linear relationship between time on the treadmill and fatigue, we then used this level of oxygen consumption to remove drift from the data of the four conditions, thus avoiding data artifacts due to randomization. Additionally, we used these data to investigate a “fatigued” condition.

Figure 1 shows a model protocol of the experiment and the relevant collected data for a single subject. Panel (a) on the top displays the raw $\dot{V}O_{2}$ and $P_{W}$ values over time. The vertical lines and alternating background colors signify the different phases of the test, as noted on the very top of Panel (a). These data rows were converted into single values by pooling over the last minute of each phase, displayed here as the horizontal bars.

In Figure 1b, we show the changes in the spatiotemporal running parameters that the subject adopted in order to fulfill the given instructions. As expected, the values changed in accordance with each other. Notably, this subject implemented the condition GCT by increasing the cadence. Other elucidations of this condition included highly elongated strides or shorter step lengths at the runner’s usual cadence.

### 2.3. Statistics

We calculated all the statistics in Python-3.6.8 using the scipy.stats-1.2.1 library. Throughout the evaluations, significance levels of *p* < 0.05 were set. We performed a repeated measures analysis of variance (ANOVA) to investigate the changes in $\dot{V}O_{2}$ in response to the varying instructions during the six conditions (baseline, SF+, SF−, arms, GCT, fatigue). We further modeled the adaptations in $\dot{V}O_{2}$ by the participants in an unambiguous way as follows.

#### 2.3.1. Oxygen Consumption during Altered Running Conditions

The raw data for $\dot{V}O_{2}$ and $P_{W}$ were recorded at 1 Hz. For each subject, we took the last 1 min of a condition to calculate the average $\dot{V}O_{2}$ and $P_{W}$ for further analysis. We determined significant changes in $\dot{V}O_{2}$ by analyzing the noise during the recording process. For this, we normalized each data row of the baseline condition (Minutes 3–5 of running) to its mean and fit a Student t-distribution to the resulting combined noise for all participants. This noise incorporated both errors stemming from the inaccuracies of the recording device, as well as fluctuations in the breathing patterns or other physiological changes. The resulting model described the variation in $\dot{V}O_{2}$ that we expected to see in every measurement. We could then compare the baseline data rows for each subject with the data for all the conditions. If more than 10% of the data points in a condition were within the 95% confidence interval (CI) of the baseline measurement, we considered this condition to not be a significantly different $\dot{V}O_{2}$. This perspicuous process corresponded exactly to a t-test for significance with *p* < 0.05 with the added benefit that we could directly mark the nonsignificant data points (cf. Figure 3, hollow points).

#### 2.3.2. Difference in Slope of Power to Oxygen

For conditions where the $\dot{V}O_{2}$ significantly differed from the baseline, we were interested in the relation between $\dot{V}O_{2}$ and $P_{W}$, as opposed to the absolute changes. We calculated the followings statistics for the slope of the relation between the change in $\dot{V}O_{2}$ and $P_{W}$ during the test conditions and compared it to the expected slope of the change in $\dot{V}O_{2}$ to $P_{W}$ during regular running (i.e., the incremental/ramp test).

Let *S* be the set of all participants in the study. For each subject *s*, the baselines ${\mathrm{VO}}_{base}^{s}$ and ${\mathrm{Pw}}_{base}^{s}$ were determined at Minute 5 of the experiment (cf. Figure 1 (left-most horizontal bar)). For altered running conditions $c_{1}\cdots c_{5}$, we calculated the relation between $\dot{V}O_{2}$ and $P_{W}$, i.e., the slope of change, as follows:(1)$${\mathrm{slope}}_{c_{i}}={\left\{\frac{{\mathrm{VO}}_{c_{i}}^{s}-{\mathrm{VO}}_{base}^{s}}{{\mathrm{Pw}}_{c_{i}}^{s}-{\mathrm{Pw}}_{base}^{s}}\right\}}_{s\in S},i\in 1\cdots 5,$$
where ${\mathrm{VO}}_{c_{i}}^{s}$ and ${\mathrm{Pw}}_{c_{i}}^{s}$ again correspond to the average of the respective range in the raw data (cf. Figure 1 (horizontal bars)). The resulting distribution $slope_{c_{i}}$ for each condition was then compared to the expected distribution of the relationship between $\dot{V}O_{2}$ and $P_{W}$ during regular running, as determined in the incremental/ramp test.

We tested the resulting distributions for normality using D’Agostino and Pearson’s omnibus test of normality [24]. We confirmed variance homogeneity between the incremental/ramp baseline and conditions using the Levene test. Lastly, we performed an ANOVA to show that there were significant differences in the expected slope and the measured results in our test conditions.

## 3. Results

### 3.1. Relation of VO${}_{2}$ and Power during the Incremental/Ramp Test

Figure 2 displays changes in $\dot{V}O_{2}$ and $P_{W}$ for all stages of the exercise test in comparison to each individual’s initial stage. A larger change in $\dot{V}O_{2}$ implies a larger change in power, and we further analyzed the slope of this relationship. Each point in Figure 2 describes the normalized average oxygen consumption and power output during the last minute of each of the stages for a single subject (cf. Equation (Equation 1)). The points in the scatter plot for Stage 1 culminate in the coordinate origin (0, 0). Linear regressions for each of the participants separately revealed a very strong linear relationship between the four different speeds and the increases in $P_{W}$ with respect to $\dot{V}O_{2}$ (*r* = 0.99, cf. light gray lines Figure 2). For each subject, the 4 Δ$\dot{V}O_{2}$
−Δ$P_{W}$ had a linear relation. The difference between the participants is expressed in the slope of these lines.

Combining the data points for all the participants also resulted in a strong correlation of *r* = 0.95 (cf. blue thick line, Figure 2). The two dashed-dotted thick dark blue lines in Figure 2 show slopes with a difference of 2·σ from the main correlation effect, i.e., the uncertainty of the slope of this regression. The likelihood for a Δ$\dot{V}O_{2}$
−Δ$P_{W}$ pair to lie outside of this cone is < 5%.

### 3.2. Running Economy

The running economy between the stages during the exercise test varied, on average, by 3.12% for each participant, whereas it varied by 6.21% for the conditions during the second part of the study. There was a significant difference between the observed variances in the two parts of our study (F(1, 60) = 39.04, *p* < 0.001).

Repeated measures ANOVA revealed a significant time effect for changes in $\dot{V}O_{2}$ (F(5, 155) = 23.51, *p* < 0.01) for the different conditions and changed running parameters. In Section 2.3, we directly test the results for each of the participants that displayed a significant difference from the baseline. Nonsignificant data points were omitted for the subsequent analysis and are displayed with hollow points in Figure 3. For the last 5 min of the running test, we observed significantly different oxygen consumption for some of the participants due to fatigue.

### 3.3. Relation between Metabolic Cost of Running and Running Power

Figure 3 demonstrates that the significant changes in $\dot{V}O_{2}$ do not imply similar changes in $P_{W}$. Each dot in Figure 3 denotes the average $\dot{V}O_{2}$ and $P_{W}$ during the last minute of each condition in the running economy test (cf. Figure 1 (horizontal bars). Statistics only include participants with a significant change in $\dot{V}O_{2}$ for each of the conditions (indicated by filled dots; the number of significant changes in the legend).

In Table 1, we show the ANOVA values for comparing the condition results vs. the expected values. For all of the conditions, there is a highly significant difference in the relation between $\dot{V}O_{2}$ and $P_{W}$. This means, there is a <0.1% probability (effectively *p* < 1.6 $\times 10{}^{-8}$) that there is *no* effective difference between the expected distribution and the measured values and that these values were observed due to noise. This same relationship is illustrated in Figure 3: The dashed-dotted cone delineates the area in the plot that all values should have fallen in, according to the incremental/ramp test. The chance for a single outlier measurement to fall outside of this cone is <5%. As athletes were independent of each other, accumulating the data yielded even the lower p-values as shown in Table 1. The trend lines in Figure 3 point at the effect strengths of these discrepancies as well.

## 4. Discussion

The goal of this study was to clarify whether it is possible to reliably measure running power using foot-worn sensors. We provide evidence for the shortcomings of these sensors when the running economy is altered. The reason for this discrepancy is that the accelerometer-based sensors approximate force applied to the ground [15], which is then used to calculate mechanical power. The energy demands on the athlete, i.e., the metabolic power, on the other hand, is the work, over time, performed in the muscle cells and cardiovascular system. During running, oxygen is consumed to contract muscles. Metabolic energy from oxygen usage is transformed into mechanical power in order to produce propulsion. The efficiency of this process is expressed in the form of RE.

Under the assumption of a fixed running economy, changes in mechanical power are directly proportional to changes in metabolic power. In the first part of our study, the combined incremental and ramp exercise test, we show that this relation holds true and validates prior work (cf. [6,9], Figure 2). Since athletes ran the exercise test in their own running style and with fairly consistent RE (cf. Section 3.2), the mechanical power measured at the foot correlates well with the metabolic power measured at the spiroergometric mask.

This assumption of a fixed RE cannot be made in practice, as it changes over time and with training [16,25,26]. By purposely altering the participants’ running economy in our experiments, we were able to show the disconnect between metabolic and mechanical power. We focused on alterations to the running form that were proven to affect the metabolic cost of running at a constant speed, i.e., the running economy. Increasing or decreasing the step frequency has been shown to significantly change the RE for experienced runners (cf. [23,27])). The same holds true for varying the ground contact time [28]. Reducing the ground contact time additionally alters both step length and frequency. Removing arm movement makes compensating for rotational motion more strenuous [16]. We reproduced the findings from prior work and showed that altering the runner’s self-chosen biomechanical parameters leads to a deteriorated running economy. At the same time, the changes in metabolic cost were not reflected in the running power measured by the foot-worn sensor. We thereby show that $P_{W}$ only correlates with $\dot{V}O_{2}$ under steady RE. It is therefore only a useful measure in situations where the athlete’s running style does not vary or change.

For single bouts of exercise, running power, as measured by foot-worn sensors, is a valid tool to keep an even effort under changing external conditions, e.g., wind or inclination [6]. Organizing long-term training using these devices cannot be expected to have the intended effects. According to the widely used model by Joyner, running performance consists of metabolic capacity, fractional usage, and RE [29]. We showed that Stryd does not measure metabolic expenditure independently of RE. A change in RE can, therefore, have arbitrary effects on the displayed power. With more training, RE and metabolic capacity at a certain pace can both change [25].

A fixed prescription of power outputs to aim for during a workout might not relate in a reasonable way to the original indicator workout: Assume an athlete improves his/her RE between Day 1 and Day 100. Running the same course in the same time on both days might yield different power readings $P_{w}^{1}$ and $P_{w}^{100}$. Since we assumed the RE to be improved, the relation between metabolic cost between Day 1 and Day 100 should be: ${\mathrm{VO}}_{2}^{1}>{\mathrm{VO}}_{2}^{100}$. As there is no well-defined correlation between $\dot{V}O_{2}$ and $P_{W}$ under a changing RE, the change between $P_{w}^{1}$ and $P_{w}^{100}$ could be zero or a change in either direction. We could expect the athlete to require less power to run the same course at the same time. Following a training schedule that increases measured running power $P_{W}$ over time can have arbitrary effects. We, therefore, do not recommend using running power for training programming, as it does not correlate with the actual metabolic cost of running when the running economy changes. Here, it might be tempting to assume the opposite and conclude that similar power outputs, but different speeds, imply an improved RE. However, our results do not allow for this deduction. We showed that the relation between $\dot{V}O_{2}$ and $P_{W}$ is arbitrary whenever the RE is changed; therefore, this comparison must be avoided.

In the existing literature, running power and economy based on foot-worn sensors have been carefully examined, but there is contradicting evidence: (1) In the accompanying white paper to their sensor by Stryd [9], power $P_{W}$ is correlated with the cost of running $\dot{V}O_{2}$. The same findings were independently validated by Cerezuela-Espejo et al. [6]. (2) A study by Muniz-Pardos et al. investigated the relation between the magnitude of acceleration and the running economy in elite athletes, finding a strong correlation of r = 0.872 [8]. The work by Austin et al. looked at the relation between power and economy as measured by a Stryd sensor, without manually altering economy [30]. (3) In a third direction of application for foot-worn sensors, Falbriard et al. investigated the feasibility of accelerometers towards determining the ground speed of the athlete, again resulting in a strong relation (accuracy = 0.00 ± 0.01 m·s${}^{-1}$, precision = 0.09 ± 0.06 m·s${}^{-1}$) [7].

Taking these prior works in combination, a sensor placed at the foot would be able to extract information about all aspects of the running form: (1) energy usage ($\dot{V}O_{2}$), (2) energy efficiency (RE), and (3) energy output (∝ speed). The information gathered from the foot of an athlete would, thereby, carry information about underlying bodily processes, from respiration to propulsion.

It is self-evident that not all of the statements can be true at the same time. In this work, we also provide evidence for contradicting statements (1): We created conditions in which $\dot{V}O_{2}$ changed significantly, but changes in power $P_{W}$ measured by foot-worn sensors did not change accordingly. We, thus, conclude that these sensors do not correlate metabolic and mechanical power, and so, metabolic running power is not measured.

Building ever-more complex models based on high-dimensional data streams introduces structural errors and logical inconsistencies as stated above. On the other hand, wearable sensors that provide a valid measurement of the metabolic cost of running in the field need to be investigated further as well. We showed that the current solutions are not sufficient to account for changes in the running conditions. This insight casts doubt over the usefulness of measuring running power using a foot-worn, accelerometer-based sensor. The promise of these sensors is to normalize running efforts between different external running conditions; instead of pace, which could be varying due to slope, running form, fatigue, and wind, power was meant to measure the actual metabolic demand on the athlete $\dot{V}O_{2}$. The sensors do not correlate with this measure. Even simple fatigue, which can be expected to occur during most training runs, resulted in changes in the demand on the athlete ($\dot{V}O_{2}$) that was not reflected in an uptick in $P_{W}$.

In this study, we examined the behavior of a single consumer product, when RE is changed. This product, the Stryd sensor, had previously been shown to provide a superior correlation under different conditions compared to other products [6]. Our aim was not to invalidate the previous studies. Instead, we demonstrated that an important factor of variance was missing in these investigations. Prior studies have not taken into account running economy. RE is an important factor in approximating metabolic cost from force measured at the athlete’s foot. The claim of foot-worn sensors for running power is that they can measure metabolic parameters under all conditions and thereby equalize between efforts. We demonstrated that there exists a variation for which this is not the case. The results of this investigation would be similar when using a different consumer product that relies on the same underlying physics.

As already pointed out by Willems [18], RE is the exact factor that determines the relation between the metabolic cost of running and the output of external power. Therefore, RE should always be included in validation studies for products that measure metabolic cost based on externally measured data. Foot-worn sensors alone do not provide a promising avenue for the development of a valid device for measuring running power. Instead, multiple sources and modalities of data should be combined, ideally from different positions of the athlete’s body (e.g., foot, wrist, and heart rate strap).

For the main part of this study, some of the participants did not have significantly different $\dot{V}O_{2}$ from their respective baselines during some of the conditions (cf. hollow points in Figure 3). The reason for this is that the participants were able to follow the instructions to varying degrees. Some participants could not alter their step frequency at all, while others had trouble matching the metronome. Compliance with the instructions was not instrumental in this experiment, as we only considered the effects of changes in the running style and not the changes themselves. Naturally, it is not to be expected that athletes alter their running form in this way during a single training run, but rather over longer periods of time. With training, athletes improve and adapt their running form to require less energy for the same speed, i.e., they improve their individual RE, over time [25]. This could be reflected in shorter ground contact time, higher cadence, or a collection of other factors [16,26].

In this study, we demonstrated the limitations of foot-worn, accelerometer-based sensors to measure metabolic running performance as expressed by $\dot{V}O_{2}$. *Running power*
$P_{W}$, as provided by foot-worn sensors, has gained popularity in recent years and was meant to help normalize between training efforts. These sensors express the running performance in Watts and supposedly correlate with the metabolic cost of running. The difference between metabolic energy consumed and mechanical energy produced, i.e., the efficiency of motion, is expressed as running economy (RE). We were able to show that altered RE leads to a disconnect between $\dot{V}O_{2}$ and $P_{W}$. Since RE changes with training and experience, we question the usefulness of foot-worn sensors for long-term programming of training.

## Figures and Tables

**Figure 1 sensors-21-04952-f001:**
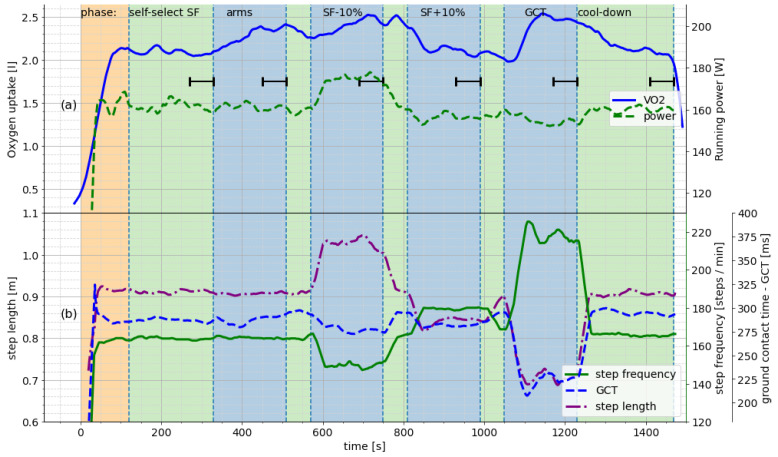
Protocol for the running economy variation test, all performed at a fixed speed. Example of a single subject. Raw values for all measurements are plotted over time (measured at 1 Hz). (**a**) Oxygen consumption and power output during the running conditions. (**b**) Spatiotemporal running parameters in reaction to instructions for the experimental conditions. $\dot{V}O_{2}$ for self-selected SF, 4 conditions and cool-down (average for last minute of each phase, cf. horizontal lines, * = significant difference): 2.06, 2.36 *, 2.49 *, 2.08, 2.47 *, 2.07. $P_{W}$: 160.62, 159.84, 170.88, 156.34, 155.62, 160.58. Best viewed in color.

**Figure 2 sensors-21-04952-f002:**
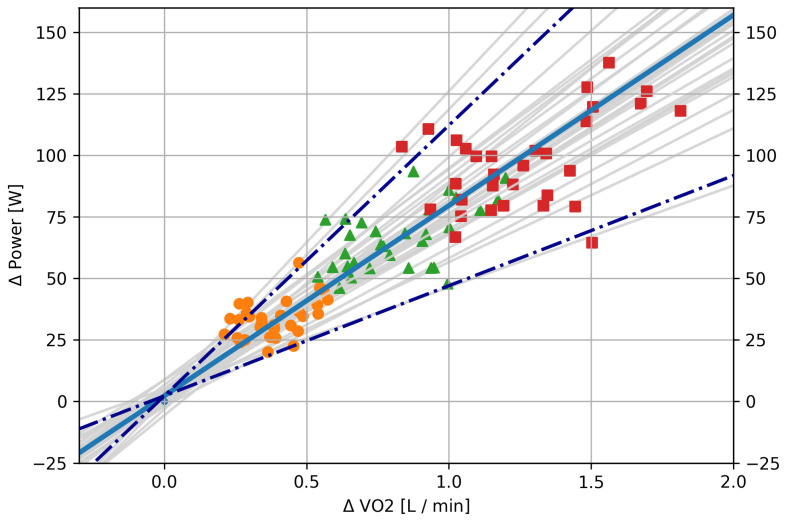
Change in respiratory $\dot{V}O_{2}$ vs. change in running power, as measured by the Stryd sensor, during the combined incremental and ramp exercise test. The first stage was used as the baseline for each of the participants (Point 0, 0). Orange circle = Stage 2; green triangle = Stage 3; red square = Stage 4. Between stages, the treadmill speed was increased by 0.5 m·s${}^{-1}$. There was a strong correlation between $\dot{V}O_{2}$ and power (cf. blue line, r=0.95). Dashed-dotted blue lines show a slope error of 2 standard deviations. Best viewed in color.

**Figure 3 sensors-21-04952-f003:**
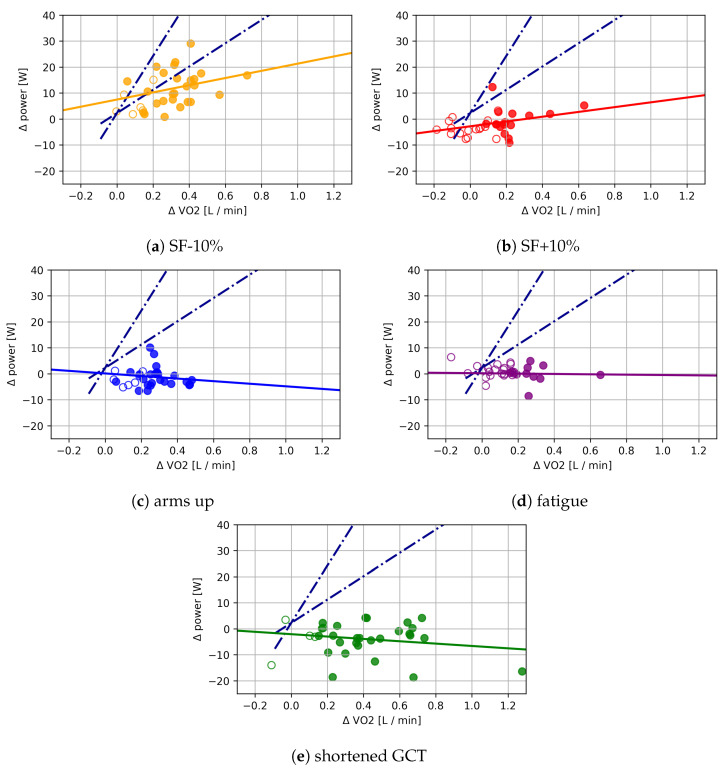
Relation of change in $\dot{V}O_{2}$ and $P_{w}$ in reaction to altered running conditions or fatigue (**a**–**e**).
The dark-blue dotted-dashed lines indicate the expected range for these points, as determined by the
incremental/ramp test. Trend lines show the main direction of a linear regression, with r and the
mean squared error (mse). Hollow points indicate a nonsignificant change in $\dot{V}O_{2}$ (the number of
participants with significant changes in $\dot{V}O_{2}$ for each condition shown in the respective legend). The
number of subjects with significant changes in $\dot{V}O_{2}$ for each condition was: (**a**) 26, (**b**) 16, (**c**) 25, (**d**)
11, and (**e**) 27.

**Table 1 sensors-21-04952-t001:** Independent ANOVA results for the relation between $\dot{V}O_{2}$ and $P_{W}$ for each condition vs. the expected distribution as determined in the incremental/ramp test. We give a visual interpretation of these results in Figure 3 as trend lines through the measured data points. Effect strengths are given as the standardized mean difference (SMD) and $\eta_{p}^{2}$ as compared to the expected baseline.

Figure 3	Condition	F	*p*	$\eta_{p}^{2}$	SMD
a	SF − 10%	36.89	<0.0001	0.24	1.15
b	SF + 10%	193.47	<0.0001	0.64	3.33
c	arms up	368.93	<0.0001	0.76	4.62
d	fatigued	160.18	<0.0001	0.61	4.98
e	GCT	429.89	<0.0001	0.78	4.72

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
