# Peer review of "Limitations of Foot-Worn Sensors for Assessing Running Power"

_sensors, 2021, doi:10.3390/s21154952_

Round 1

Reviewer 1 Report

The study results are well presented. I have one minor comment, authors can provide a brief introduction to some terms which can save some reader's effort to search through the reference/web. For example, metronome, stryd sensor, etc.

Author Response

Dear Reviewer 1,

The study results are well presented. I have one minor comment, authors can provide a brief introduction to some terms which can save some reader's effort to search through the reference/web. For example, metronome, stryd sensor, etc.

Thank you very much for your valuable feedback. We added some more details about the Stryde sensor to the Methods section (purple highlight in revision): “This sensor is attached to the athlete's shoe via a clip into the shoelaces and connected to a watch via Bluetooth. It is lightweight (8.5g), unobtrusive (4cm length), and does not impact the athlete's running [15]”

Generally, we try to go into technical detail, where necessary. For some products, an additional look at the company’s website is advised to get a feel for what the product looks like, what it is advertised towards, and how it is meant to be used. This gives the reader a deeper understanding than we could provide. We list hyperlinks to facilitate and encourage this background reading.

Best,

Tobi

Reviewer 2 Report

This study discussed whether running power as measured by foot-worn sensors is considered to be associated with the metabolic cost of running. The experimental results stand in contrast to the previously reported link between metabolic and mechanical running characteristics estimated by foot-worn sensors. This casts doubt on the feasibility of measuring running power in the field, as well as using it as a training signal. However, there are some concerns need to be addressed as follows:
1. In line 3, “32 experienced participants (age = 28 ± 7 years, weekly running distance = 51± 24)”, lack of units for weekly running distance. I think the unit is km.
2. I want to know what the sampling rate of each machine and device is? How to synchronize them?
3. In line 118, The author uses Stryd sensor as foot-worn sensors to collect data. Is it possible that Stryd sensor is a more inaccurate sensor? Have you considered trying several foot-worn sensors for experiments and comparison? This is a big problem.
4. In section Discussion, there are differences between your research results and most previous research results. Can you discuss this point and where are the possible problems?
5. The English grammar of this paper should be improved. I recommend to use the English editing service to polish the sentences.

Author Response

Dear Reviewer 2,

This study discussed whether running power as measured by foot-worn sensors is considered to be associated with the metabolic cost of running. The experimental results stand in contrast to the previously reported link between metabolic and mechanical running characteristics estimated by foot-worn sensors. This casts doubt on the feasibility of measuring running power in the field, as well as using it as a training signal. 

Thank you very much for your insightful and valuable feedback! We added your feedback to our manuscript and highlighted changes inspired by your notes in green. Please find below a more detailed discussion of all of your comments.

However, there are some concerns need to be addressed as follows:

  1. In line 3, “32 experienced participants (age = 28 ± 7 years, weekly running distance = 51± 24)”, lack of units for weekly running distance. I think the unit is km.

Yes, you are of course absolutely right here. Later, in the methods section, we mention the units but forgot to do so in the abstract. Thanks for catching this mistake.

  1. I want to know what the sampling rate of each machine and device is? How to synchronize them?

This is a very valid point and indeed a potential source of error. We added a paragraph in the methods section to go into details about sampling rate, synchronization, and how we get around the need for perfectly synchronous data.

  1. In line 118, The author uses Stryd sensor as foot-worn sensors to collect data. Is it possible that Stryd sensor is a more inaccurate sensor? Have you considered trying several foot-worn sensors for experiments and comparison? This is a big problem.

Thank you very much for raising this issue. Our aim is to show inaccuracies in the validation method of previous studies instead of a specific device. We show that using the previously accepted methods of validation do not cover all sources of variation and particularly misses changes due to running economy. We use the sensor that scored best in the previous validation studies to demonstrate that the validation process was flawed. 

Adding more sensors on the foot or processing the signal differently will not change the results of our investigation. In this work, we show that there are fundamental issues with measuring respiratory indicators using an accelerometer attached to the athlete’s foot. We did not concisely prove that this is completely impossible. To do so, we agree, that we would indeed need to test every available product for foot-based running power. Better yet, we would design that experiment in a way, where we use raw data from accelerometers and measure the mutual information with spirometric data.

Instead, the focus of the present study was to show that the notion of foot-based running power as it is currently used in consumer products does not relate to metabolic power in the way that it is advertised. The correlation between VO and Pw has been independently validated [6] and Stryd has been identified as the best predictor for metabolic power (out of 2 foot-worn products). In order to break this pro-claimed correlation, we show that the best-in-class product behaves arbitrarily when RE is changed. 

Assume that the runner-up product from their validation study (RunScribe) would have been able to account for Running Economy. This would imply, that a method M that factors RE into the relation between VO and Pw results in a weaker overall correlation between VO and Pw. This discrepancy would again be evidence for the fact that VO and Pw do not carry equivalent information; Method M would need to be some function that mitigates the difference, meaning that RE plays a factor in the relation. The relation of VO and Pw is dependent on RE. This is the exact statement of our study.

  1. In section Discussion, there are differences between your research results and most previous research results. Can you discuss this point and where are the possible problems?

Prior validation studies had not considered RE at all when investigating the relation between VO and Pw, which is why our results differ from the literature. As we mention in the text, changes in RE are not to be expected over a single bout of exercise, but rather over a longer period of time. Previous research results only investigated the correlation of data for changing conditions that are to be expected for a single run. We manipulate the RE in our experiments and thereby get a hint at the expected changes when using these products over a longer period of time.

  1. The English grammar of this paper should be improved. I recommend to use the English editing service to polish the sentences.

Thanks! The paper underwent an additional thoroughly conducted grammar, wording, and typo check.

I hope this clarifies our study design a bit more and convinces you that the result is just as strong using only this single consumer product.

Thank you,

Tobi

Reviewer 3 Report

The reviewed manuscript sought to assess the validity of a foot-worn running power meter. By varying running economy through purposeful manipulation of spatio-temporal gait characteristics, the authors show a significant effect on metabolic cost that was not detected by the foot-worn device. These negative findings are in contrast to previous reports in the field, and cast doubt on the utility of foot-worn power monitoring as a means to accurately reflect the metabolic cost of running. Overall, I found the paper to be well-written, although a bit wordy at times, the experimental design appropriate, and the conclusions in-line with the observed data. I ask the authors to please consider the following, which I feel may help to improve the quality of their work.

 Introduction

  • The introduction is excessively long and should be condensed. The second paragraph in particular seems out of place and moderately redundant with the last paragraph (lines 66-75).

Methods

  • Was there any control of recent injury or other factors that might affect gait mechanics?
  • Please describe in greater detail how the aerobic threshold was identified
  • Was there any control for footwear of the athletes?
  • Were the conditions randomized or did the athletes all proceed through in the same sequence?

Results

  • Figure 2: Should the units on the x-axis by “L/min”? Same goes for Figure 3.

Discussion

  • The first sentence of the paragraph starting “For a single bout of exercise..” (lines 284-286) does not make sense
  • In the paragraph 292-303 the authors use their data to make a strong case against the prescription of running workouts using power. While I fully agree, I do wonder if the “power” metric provided by devices such as the Styrd might still be useful. Specifically, if an athlete were to run at the same power on day 1 and day 100 of a training program but improve their speed on day 100 would that not allude to an improvement in running economy? This might also provide valuable information to coaches and athletes, though admittedly this is not suggested/intended use of such a device.

Author Response

Dear Reviewer 3,

The reviewed manuscript sought to assess the validity of a foot-worn running power meter. By varying running economy through purposeful manipulation of spatio-temporal gait characteristics, the authors show a significant effect on metabolic cost that was not detected by the foot-worn device. These negative findings are in contrast to previous reports in the field, and cast doubt on the utility of foot-worn power monitoring as a means to accurately reflect the metabolic cost of running. Overall, I found the paper to be well-written, although a bit wordy at times, the experimental design appropriate, and the conclusions in-line with the observed data. I ask the authors to please consider the following, which I feel may help to improve the quality of their work.

Thank you very much for your insightful and valuable feedback! We added your feedback to our manuscript and highlighted changes inspired by your notes in blue. Please find below a more detailed discussion of all of your comments.

Introduction

The introduction is excessively long and should be condensed. The second paragraph in particular seems out of place and moderately redundant with the last paragraph (lines 66-75).

The introduction is admittedly somewhat long-winded. We wanted to pay tribute to the complexity of the topic and motivate reasons why these sensors exist, why there is a need for them, what has been tried before, and why prior work has not considered the manipulations we are testing in this study.

In the paragraph you reference, we give a brief overview of the results of the paper to create the silver lining in our argument.

Methods

Was there any control of recent injury or other factors that might affect gait mechanics?

Please describe in greater detail how the aerobic threshold was identified

Was there any control for footwear of the athletes?

Were the conditions randomized or did the athletes all proceed through in the same sequence?

Thank you for pointing out these shortcomings. We added some more details in the Methods section to remedy these questions.

Results

Figure 2: Should the units on the x-axis by “L/min”? Same goes for Figure 3.

Indeed. Great catch! Thanks for pointing this out.

Discussion

The first sentence of the paragraph starting “For a single bout of exercise..” (lines 284-286) does not make sense

In the paragraph 292-303 the authors use their data to make a strong case against the prescription of running workouts using power. While I fully agree, I do wonder if the “power” metric provided by devices such as the Styrd might still be useful. Specifically, if an athlete were to run at the same power on day 1 and day 100 of a training program but improve their speed on day 100 would that not allude to an improvement in running economy? This might also provide valuable information to coaches and athletes, though admittedly this is not suggested/intended use of such a device.

We really enjoyed your discussion of the potential applications to the newly gained insights from this study. Thank you for these ideas. Unfortunately, things aren’t as cut and dry as one might hope.

In this work, we show that there is an arbitrary relation between running power (measured by Stryd) and true metabolic cost when changing the running economy. This means that whenever the RE changed, we cannot interpret the power values that come from the device. Thus, it would be difficult to judge whether a change in RE was the only difference between day 1 and 100. 

Consider the extreme scenario where half of the speed improvement is due to RE and the other half due to an increased power output during the time trial on day 100. This means our athletes improved both their capacity to perform work (and expand energy) for a duration of time, as well as decrease the amount of energy consumed for propulsion. The results of our study suggest that it is impossible to disentangle these two factors from just the single value measured at the foot. It could happen that due to changes in RE, the power is underreported in a way that makes it seem like the power on day 1 and 100 are the same. In fact the metabolic power could have changed in either direction and the sensor is not able to pick this up.

In the cited work by Muniz-Pardos from 2018, they use foot-worn sensors to measure RE itself. While we did not reproduce and investigate these results, we suspect that similar problems will arise when trying to derive RE itself from an accelerometer worn at the foot. It seems counterintuitive that information about respiration would be interpolated from the foot.

Thank you,

Tobi

Round 2

Reviewer 2 Report

This study discussed whether running power as measured by foot-worn sensors is considered to be associated with the metabolic cost of running. The experimental results stand in contrast to the previously reported link between metabolic and mechanical running characteristics estimated by foot-worn sensors. This casts doubt on the feasibility of measuring running power in the field, as well as using it as a training signal. Authors almost and completely answered all questions. However, there are some format mistakes needed to correct.
1. In line 56, the formula should be “F = ma”, not “F = maa”.
2. There must be a space between the number and the unit. For example:
In line 80, 13.1ml => 13.1 ml
In line 126, 8.5g => 8.5 g, 4cm =>4 cm
In line 135 & 137, 1Hz => 1 Hz
Others.
3. In whole manuscript, et. al => et al.
4. Caption of "Table 1. Independent ANOVA results..." should be above the table, not below.

Author Response

Dear Reviewer 2,

This study discussed whether running power as measured by foot-worn sensors is considered to be associated with the metabolic cost of running. The experimental results stand in contrast to the previously reported link between metabolic and mechanical running characteristics estimated by foot-worn sensors. This casts doubt on the feasibility of measuring running power in the field, as well as using it as a training signal. Authors almost and completely answered all questions. 

Thank you very much for your concise and informed summary of our work. We want to thank you again for the points you raised in your first round review and believe integrating your feedback made this work a stronger contribution. In the following, we address your corrections item-by-item but abstained from highlighting the changes in the revision of the manuscript.

However, there are some format mistakes needed to correct.

1. In line 56, the formula should be “F = ma”, not “F = maa”.

Indeed, the correct version of Newton’s second law only contains acceleration once. Thank you for pointing out this error! If force was quadratic in acceleration, the measured effects might have been even stronger.

2. There must be a space between the number and the unit. For example:

In line 80, 13.1ml => 13.1 ml

In line 126, 8.5g => 8.5 g, 4cm =>4 cm

In line 135 & 137, 1Hz => 1 Hz

Others.

Thank you for looking into our numbers in this great detail. We carefully went through the entire work again and made sure that our formatting is more consistent.

3. In whole manuscript, et. al => et al.

Thanks! We updated our macro for et al. to the correct formatting. 

4. Caption of "Table 1. Independent ANOVA results..." should be above the table, not below.

The caption for the table is now in the correct position.

Again, thank you for your detailed review of our manuscript!

Best,

Tobias Baumgartner on behalf of all co-authors